# Instance Segmentation of Multiple Myeloma Cells via Hybrid Task Cascade

## Abstract

Multiple Myeloma (MM) is a blood cancer that develops when plasma cells expand abnormally in the bone marrow. Early detection of MM is beneficial for accurate treatment in time and draws increasing recognition. There are several endeavors to construct computer-assisted automatic diagnostic tools for myeloma cell detection. Of late, deep learning based methods have been expected to apply for the detection and segmentation of cells of interest from microscopic images. In this paper, an ensemble framework is designed for the detection and segmentation of myeloma cells. Extensive experimental results on the SegPC-2021 Challenge dataset demonstrate that the proposed method achieves promising performance. Notably, our overall framework obtains 93.72 mIoU on the test dataset of the challenge phase-2 and exceeds the 1st place record of the leaderboard.

**Keywords:** Instance Segmentation, Deep Learning, Multiple Myeloma.

## 1 Introduction

MM is a blood cancer caused by the accumulation of irregular plasma cells in the bone marrow. It is thought to be highly curable if MM can be detected in the early stages. However, the diagnosis of MM is a subjective and time-consuming task (Saeedizadeh et al., 2016). The common method of identification of myeloma cells is carried out by using microscopic analysis based on their histologic and cytologic features (Tehsin et al., 2019). Recently, it is expected to build computer-assisted diagnostic tools to assist pathologists to automatically detect myeloma cells from microscopic images, which help to reduce the diagnostic time and improve the accuracy of identification (Tehsin et al., 2019).

Instance segmentation is a basic computer vision task that localizes objects of interest and predicts pixel-level segmentation for each instance. The task of building the computer-assisted tools for MM is closely related to instance segmentation because it integrates detection, classification and segmentation of specific objects. Achieving accurate and robust instance segmentation of myeloma cells is difficult due to the following challenges:

- Both cytoplasm and nucleus are required to segment for each instance of myeloma cells.
- The amount of nucleus is different for each cell.
- The cells are either clustered or isolated.
- There are three cases for the clustered cells: nucleus-nucleus cluster, nucleus-cytoplasm cluster, and cytoplasm-cytoplasm cluster.
- The cytoplasm of a cell can be close to the microscopy image's background, making it difficult to distinguish and segment the boundary of the cell.

For instance segmentation, there are two main types of methods, single-stage methods (Liu et al., 2016; Redmon et al., 2016; Wang et al., 2017) and multi-stage methods (Ren et al., 2017; Dai et al., 2016; Cai & Vasconcelos, 2021). Generally, multi-stage methods are more accurate but slower than single-stage methods. It is usually the case that multi-stage methods (He et al., 2017) achieve better performance when speed and memory are not critical. In this term, accuracy is more important than inference speed. Therefore, our proposed framework is essentially a multi-stage method and we show comparisons with other different multi-stage methods.

## 2 Related Work

Instance segmentation of cells from microscopic images is a relatively new research topic and there are several works in recent years (Vyshnav et al., 2020). However, there are few applications in

MM specific tasks. Saeedizadeh (2016) propose the first work on MM detection and classification. Thresholding is used to detect myeloma cells. The nucleus and cytoplasm are then split using bottleneck and watershed algorithms. Finally, an SVM classifier is applied to classification on features extracted from the cytoplasm and nucleus. Meanwhile, Gupta et al. (2017, 2018), propose a stain color normalization for microscopic image and it is applied to PCSeg – a MM-specific automatic segmentation tool. Tehsin et al. (2019) use CNN model Alexnet to extract features from microscopic images and then use SVM for the classification of myeloma cells.

It is clear that most of the related researches are based on conventional image processing methods. Currently, the only work on MM detection and segmentation using deep learning based methods is performed by Vyshnav et al. (2020). It shows that Mask-RCNN obtains excellent performance which is much better than previous methods. It can handle the majority of the problems related to myeloma cell segmentation. But it misses some of the understained cells and misclassifies some normal cells. Moreover, in the past several years, there are much remarkable progress in object detection and instance segmentation. Those advanced methods are expected to to conduct the detection of myeloma cells to obtain further improvement and higher accuracy.

Mask R-CNN (He, 2017) is arguably the most popular instance segmentation method. The Mask-RCNN extends an extra mask head on Faster R-CNN (Ren, 2017) to predict object mask and it is the cornerstone of modern instance segmentation. Cascade Mask R-CNN (Cai & Vasconcelos, 2021) shows that cascade is a strong architecture, that can boost performance to the instance segmentation task. Furthermore, Hybrid Task Cascade (Chen et al., 2019) is a stronger variant on Cascade Mask R-CNN. It uses a semantic segmentation branch to provide spatial context and interleaves detection and segmentation features for joint multi-stage processing. It ranks 1st on the COCO challenge 2018 which is SOTA for instance segmentation. In this paper, we adopt HTC as our baseline method. Furthermore, DetectoRS (Qiao et al., 2020) proposes Recursive Feature Pyramid (RFP) and Switchable Atrous Convolution (SWC) and combines them in the backbone design for object detection. It can significantly improve the performance of the HTC.

## 3    Method

### 3.1    Two Branches

Since the cytoplasm may be similar to the microscopy image's background, it is difficult to segment the boundary between the nucleus and cytoplasm. The shape of the nucleus is usually round or oval while the shape of the cytoplasm sometimes is irregular. In order to get better segmentation between the nucleus and cytoplasm, we use two branches to segment instances of the two classes, as shown in Figure. 1.

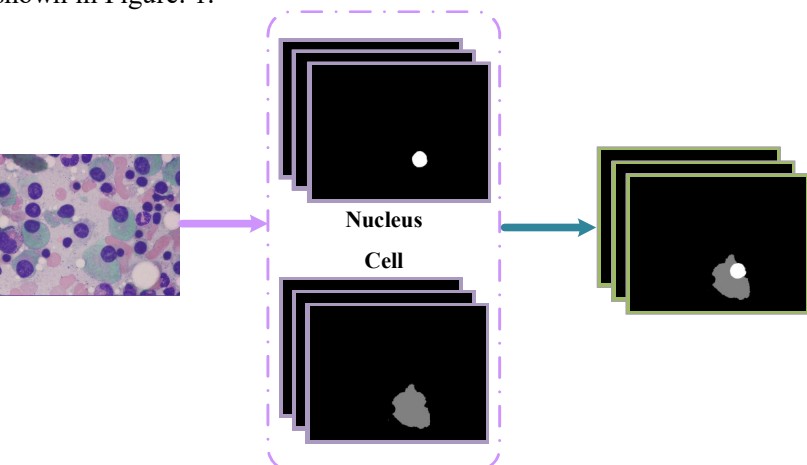

**Figure 1.** Two branches to segment instances of the cell..

Branch 1 is used to segment the whole cell, that includes the nucleus and cytoplasm. Branch 2 is used to segment the nucleus only. Then we can get masks of the cytoplasm from masks of cell subtracting masks of the nucleus, which are better than segmentation of the nucleus and cytoplasm respectively.

## 3.2   Hybrid Task Cascade

Hybrid Task Cascade (HTC) (Chen et al., 2019) is a excellent instance segmentation method and it is one of the latest families of Mask-RCNN (He, 2017). HTC integrates the previous stage's mask features into the current stage, improving the information flow among mask branches. Figure.2 shows the architectures of the proposed MM instance segmentation using HTC.

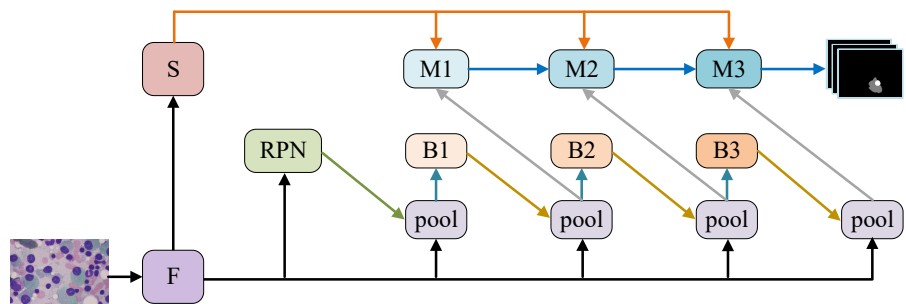

**Figure. 1.** The architecture of HTC. "F" is the backbone network, "pool" represents the Region of Interest Align layer (ROI Align), "S" is a semantic segmentation head. "B1" and "M1" indicate the bounding box head and mask head of the 1st stage, and so on.

Initially, a specific backbone network extracts feature maps from input microscopic images. Then, Region Proposal Network (RPN) (Ren et al., 2017) generates object proposals from the feature maps. Region of Interest Align layer (Ren et al., 2017) resizes these proposals to the same size as the feature maps. These proposals are processed by the detection sub-network ("B1"), denoted as the bounding box head 1, to create the 1st bounding box predictions. These bounding box predictions are passed to the next ROI Align layer and then the segmentation sub-network ("M1"), denoted as the mask head 1, which creates the 1st mask predictions from the 1st bounding box predictions. Meanwhile, the 2nd bounding box predictions are created by the "B2" and so on. Since we adopt three-stage cascade structure, the "M3" creates the final mask predictions.

Moreover, there is a direct information flow between mask heads at each stage, further improving the accuracy of mask predictions. With this direct information flow, mask features in each stage are brought into direct interaction instead of being isolated. To further identify the foreground from the cluttered background, a semantic segmentation head ("S") is added to predict pixel-wise semantic segmentation, which get spatial contexts from the whole image. It is strongly complemented with the bounding box heads and mask heads and benefits higher accuracy by distinguishing the foreground from the cluttered background.

## 3.3   Ensemble Models.

We adopt Feature Pyramid Networks (FPN) (Lin et al., 2017) with ResNeXt-101-64x4d (Xie et al., 2017) as the backbone of HTC, denoted as Model 1. We incorporate Recursive Feature Pyramid (RFP) (Qiao, 2020) and Switchable Atrous Convolution (SAC) (Qiao, 2020) with ResNet-50 as the backbone of HTC, denoted as Model 2.

Since the segmentation of the whole cell is more difficult than the nucleus. We only use Model 1 to predict the instances of the nucleus in Branch 2. And we ensemble Model 1 and Model 2 with non-maximum suppression(NMS) to predict the instances of cells including the nucleus and

cytoplasm. For Branch 1, it requires higher precision and it determines most of the score of evaluation. Finally, the flowchart of our ensemble framework is shown in Figure. 3.

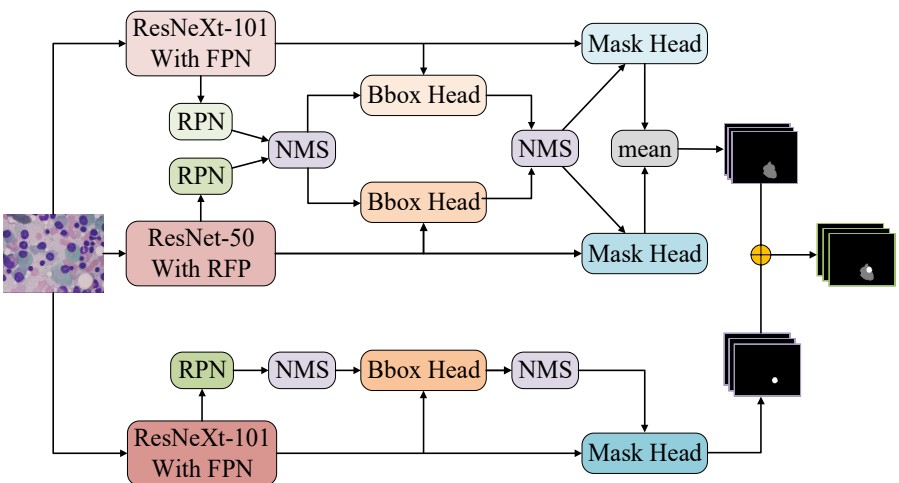

**Figure. 2.** The flowchart of the proposed ensemble framework.

## 4    Experiment

Experiments are carried out on the SegPC-2021 Challenge dataset. Models are trained on the train set (208 images) and validated on the validation set (90 images). Then we test the models on the test dataset of challenge phase-2 which contains 200 images.

For ablation studies, all methods are implemented with PyTorch and the excellent MMDetection (Chen et al., 2019) framework. We train models on a single NVIDIA Tesla V100 graphics card and the batch size is 2. Input images are resized to $1066 \times 800$. We apply light augmentations to the original image. It includes horizontal flip and random rotate 90 degrees with the probability of 0.5 both. We train models for 24 epochs with 1000 iterations of warming up. We set the initial learning rate as 0.01, then reduce it by 0.1 after 20 and 23 epochs respectively. Stochastic gradient descent (SGD) is used as the optimizer where momentum is 0.9 and weight-decay is 0.0001. For testing, we set the IoU threshold as 0.5 for NMS and set the score threshold as 0.05 for inference.

### 4.1    Datasets

The datasets with annotations are provided by SBILab, the organizer of the SegPC-2021 Challenge (Gupta et al., 2021). Microscopic images are captured from bone marrow aspirate slides of the multiple myeloma patients. These slides are then stained with Geometry-inspired chemical and tissue invariant stain normalization (GCTI-SN) (Gupta et al., 2020). Two cameras are used to capture microscopic images in raw BMP format: (1) Olympus cellSens software at a resolution of $2040 \times 1536$ pixels. (2) Nikon camera at a resolution of $1920 \times 2560$ pixels.

The training dataset consists of a total of 298 images and the test dataset consists of a total of 200 images. For each image of the training dataset, annotations corresponding to the cell of interest have been provided. We split the training dataset in the ratio of 70/30 into train set (208 images) and validation set (90 images). A sample microscopic image with annotations is shown in Figure.4. The box and mask heads are supervised by the official annotations and the semantic head is supervised by annotations created at our end.

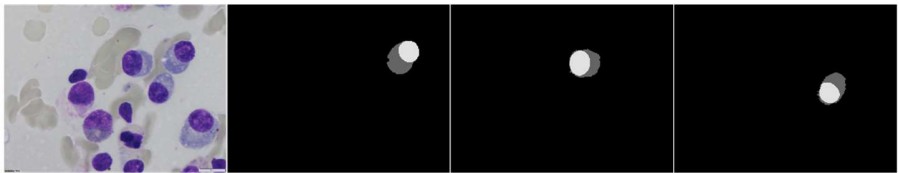

**Figure. 3.** A sample image with annotations of dataset.

## 4.2    Evaluation Metric

We evaluate and choose our best models based on the standard mean average precision (mAP) metric which calculates average APs across eleven IoU thresholds from 0.5 to 0.95 with 0.05 interval. Another evaluation metric is mean intersection over union (mIoU) which IoU will be calculated for each instance of the cells of interest. The mAP is used to measure the performance of detection while the mIoU is used to measure the performance of segmentation.

## 4.3    Ablation Studies

Table. 1 shows the impact of different backbones on the performance. And it also shows the comparison of HTC with other instance segmentation methods. For fair comparisons, FPN is used in the backbones of all models.

| Method | Backbone | mAP | mIoU |
|---|---|---|---|
| Mask R-CNN | ResNet-50 | 75.8 | 90.90 |
| Mask R-CNN | ResNet-101 | 76.8 | 91.74 |
| Cascade Mask R-CNN | ResNet-50 | 77.2 | 91.72 |
| Cascade Mask R-CNN | ResNeXt-101-64x4d | 77.7 | 92.02 |
| Hybrid Task Cascade | ResNet-50 | 76.9 | 92.31 |
| Hybrid Task Cascade | ResNeXt-101-32x4d | 77.9 | 92.34 |
| Hybrid Task Cascade | ResNeXt-101-64x4d | 78.3 | 92.76 |

**Table 2.** Comparison of SOTA instance segmentation methods on the test dataset.

| Method | mAP | mIoU |
|---|---|---|
| HTC baseline | 78.3 | 92.76 |
| +DCN | 78.5 | 92.96 |
| +Muti-scale | 79.3 | 93.34 |
| +DIoU loss | 79.7 | 93.47 |
| +Soft-NMS | 80.2 | 93.50 |
| +TTA | 80.7 | 93.60 |
| +ensemble | 82.0 | 93.72 |
| Overall framework | **82.0** | **93.72** |

**Table 2.** Step-by-step improvements of adopting different modules.

## 4.4 Bells and Whistles

Our overall framework obtains 82.0 mAP and 93.72 mIoU on the test dataset of the challenge phase-2. Here we list the step-by-step improvements of various modules adopted in our methods to obtain the performance. Results are shown in Table. 2.

- **HTC Baseline.** We adopt ResNeXt-101-64x4d with FPN as the backbone. The baseline achieves 78.3 mAP and 92.76 mIoU.
- **DCN.** We adopt deformable convolution (DCN) (Dai et al., 2017) from the 3rd stage to the last stage (5th stage) of the backbone.
- **Multi-scale Training.** We adopt multi-scale training. The long edge of the image is set to 1600 while the short edge is randomly chosen from 400 to 1200.
- **DIoU Loss.** We adopt Distance-IoU loss(DIoU loss) (Zheng et al., 2020) to replace the original Smooth L1 loss for the bounding box regression.
- **Soft-NMS.** We apply Soft-NMS (Bodla et al., 2017) to bounding box fusion. Soft-NMS yields higher mAP but produces numerous predictions.
- **Test-time Augmentation.** For testing, we use 3 scales and horizontal flip and then merge the results together. The scales are 533×400, 1440×1080, 1600×1200.
- **Ensemble.** We use the proposed ensemble network in Section 3.3.

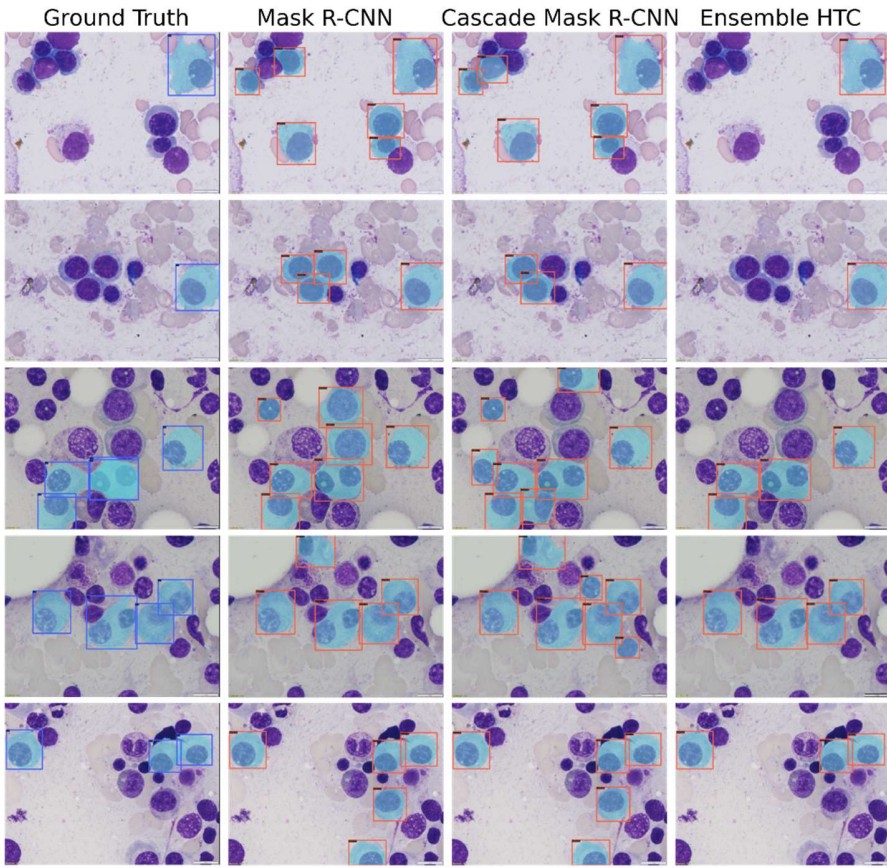

**Figure. 5.** From left to right: visualization of the results by the ground truth, Mask-RCNN, Cascade Mask-RCNN and ensemble HTC(ours).

### 4.5 Visualization

Figure. 5. provides visualizations of the myeloma cell predictions for different methods. From left to right, the results are from Mask-RCNN, Cascade Mask-RCNN, and ensemble HTC(ours) where mAP is 75.8, 77.7, and 82.0 respectively. We only show segmentations of Branch 1, e.g. segmentations of whole cell, where masks are colored by blue. We can see that our ensemble framework obtains better accuracy for the detection and segmentation of myeloma cells.

## 5 Conclusion

Early detection of Multiple Myeloma draws increasing recognition and efforts are underway to construct computer-assisted automatic diagnostic tools for myeloma cell detection. Such computer-assisted tools require accurate detection and segmentation of myeloma cells. In this paper, An ensemble framework is designed for myeloma cell instance segmentation. We use two branches to segment instances of cytoplasm and nucleus. We adopt HTC as our baseline method and use stronger backbones, effective modules, and ensemble strategies. We obtain a promising result of 82.0 mAP and 93.72 mIoU on the SegPC-2021 Challenge dataset.

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
