# OpenReview forum: "Instance Segmentation of Multiple Myeloma Cells via Hybrid Task Cascade"
_MICCAI.org/2021/Workshop/COMPAY — Reject_

### Official Review · Reviewer_AcDd · 2021-08-17
**seems promising, improves sota on the evaluated task**

**Rating:** 6
**Confidence:** 3

**Review:**

The authors propose an ensemble framework for the detection and segmentation of myeloma cells. The main novelty is the usage of two branches for the nucleus and cytoplasm. The proposed ensemble approach improves the mAP from 78.3 (baseline) to 82.
This work seems to include (and depend) on a lot of technical details and hyperparameters, I would suggest making the code public for reproducibility.

---

### Official Review · Reviewer_hpxV · 2021-08-19
**Combining existing approaches to achieve SOTA performance on a public test set of myeloma cell segmentation**

**Rating:** 5
**Confidence:** 3

**Review:**

This paper presents a two-branch architecture for the segmentation of myeloma cells in cytology images, where the two branches are used to segment the nucleus and the cytoplasm separately, to be then combined.
The model is based on previous work (HTC, Chen et al., 2019), and the proposed solution seems to be effective, outperforming the state of the art on the test set of the public SegPC-2021 challenge.

The main novelty of the paper is in the use of multiple branches to segment two different parts of the cell, and the combination of bounding box predictions in model 1 and model 2, while the rest relies on existing previous work.

A good aspect of the present method is that it achieves SOTA performance on a public test set.

On the other hand, some aspects of the paper are not completely clear:
* In Figure 2, it is not clear what is model 1 and model 2, and which blocks implement the HTC architecture presented in Figure 1.
* The paper lacks a discussion section, it does include a Visualization section that refers to Figure 5, and a "Bells and Whistles" section, which provides some details about features adopted to achieve the presented results, but does not add insights on the presented methods.
* The use of English or sentence construction could be improved, e.g., "is a excellent", "may be similary to the microscopy", "Since the segmentation of the whole cell is more difficult than the nucleus. We only..."

---

### Official Review · Reviewer_Gjb5 · 2021-08-23
**Interesting application of instance segmentation to MMC detection that leaves open several questions.**

**Rating:** 5
**Confidence:** 4

**Review:**

This paper describes the application of an deep instance segmentation method to the detection and segmentation of multiple myeloma cells (MMC), based on images and annotations provided by the SegPC-2021 Challenge. The authors use a Hybrid Task Cascade model, a new member of the Mask-RCNN family to train a model for the segmentation of both the nuclei and cytoplasm of MMCs. The main contribution of the paper is the proposal of an ensemble framework with separate models trained for the segmentation of the cytoplasm. The authors present results, including an ablation study, that show the proposed ensemble approach outperforms other approaches.

The paper describes the study in sufficient detail, although some questions around the input images and annotations remain (esp. for readers not familiar with SegPC-2021): What staining is used? What is the resolution/mpp? Moreover, details on the self-made annotations for the semantic head are missing. How are these generated, automatically or manually by a certified pathologist?

More importantly, the argumentation for the need and details for the ensemble (which is the main contribution) are not made clear enough. For instance, why these specific models (ResNet50 and ResNeXt101)? Why was a ensemble only used for the cytoplasm segmentation model?  How was the “mean” of two binary masks used exactly (as “and” or “or” operation)?

As I understand, the key diagnostic metric is the cell count of MMCs in an image.  Therefore, a medically relevant addition to the quantitative mAP and mIuO results presented in tables 1 and 2 would be the number of false positives and false negatives. Also, the evaluation in terms of recognition quality, segmentation quality and panoptic quality would be interesting for benchmarking.

The readability of the manuscript can be improved and would benefit from being edited by a native English speaker. Also, the numbering of the figures and tables is wrong and should be corrected.

---

### Decision · Program_Chairs · 2021-08-25

Reject